# The Risk-Reducing Effect of Aspirin in Lynch Syndrome Carriers: Development and Evaluation of an Educational Leaflet

*Rajneesh Kaur,\* Cassandra McDonald, Bettina Meiser, Finlay Macrae, Sian K Smith, Yoon Jung Kang, Michael Caruana, and Gillian Mitchell*

Carriers of germline mutations in genes associated with Lynch syndrome are at increased risk for colorectal, endometrial, ovarian, and other cancers. There is evidence that daily consumption of aspirin may reduce cancer risk in these individuals. There is a need for educational resources to inform carriers of the risk-reducing effects of aspirin or to support decision-making. An educational leaflet describing the risks and benefits of using aspirin as risk-reducing medicine in carriers of Lynch-syndrome-related mutations is developed and pilot tested in 2017. Carriers are ascertained through a familial cancer clinic and surveyed using a mailed, self-administered questionnaire. The leaflet is highly rated for its content, clarity, length, relevance, and visual appeal by more than 70% of the participants. Most participants (91%) report "a lot" or "quite a bit" of improvement in perceived understanding in knowledge about who might benefit from taking aspirin, its benefits, how long to take it, the reduction in bowel cancer risk, and the optimal dosage. A few (14%) participants seek more information on the dosage of aspirin. This leaflet will be useful as an aid to facilitate discussion between patients and their health care professionals about the use of aspirin as a risk-reducing medication.

## 1. Introduction

Lynch syndrome, previously known as hereditary non-polyposis colorectal cancer, affects ≈1 in 280 men and women.[1] Individuals with mutations in mismatch repair genes (MMR)—MutS Homolog 2 *(MSH2), MutS homolog (MSH6), MutL homolog 1 (MLH1),* and PMS1 Homolog 2, Mismatch Repair System Component *(PMS2)*—incur up to a 70% lifetime risk of developing colorectal cancer (CRC) with gene-specific mutations.[2] CRC is the most common cancer that occurs in people with Lynch syndrome,[3] and Lynch syndrome is the cause of 3% of all CRCs.[4] In addition, women with Lynch syndrome are also at increased risk of developing endometrial and ovarian cancer.[5–7] Lynch syndrome incurs a between 26% and 43% lifetime chance of developing endometrial cancer[8] and an up to 13-fold lifetime chance of developing ovarian cancer.[8] In comparison to the general population,

R. Kaur
Medical Education Office
UNSW Sydney
New South Wales, Australia
E-mail: rajneesh.kaur@unsw.edu.au

R. Kaur
Medical Education Office
The University of Sydney
Edward Ford Building, Sydney, New South Wales 2006, Australia

C. McDonald
The Kinghorn Cancer Centre
St Vincent Hospital
Victoria Street, Darlinghurst, New South Wales 2010, Australia

B. Meiser, S. K Smith
Psychosocial Research Group
UNSW Sydney
High Street, Sydney, New South Wales 2052, Australia

F. Macrae
Department of Colorectal Medicine and Genetics
and Department of Medicine
The Royal Melbourne Hospital
University of Melbourne
Parkville, Victoria 3010, Australia

Y. J. Kang, M. Caruana
Daffodil Centre
University of Sydney
Sydney, New South Wales 2006, Australia

G. Mitchell
Familial Cancer Centre
Peter MacCallum Cancer Centre
Parkville, Victoria 3010, Australia

G. Mitchell
The Sir Peter MacCallum Department of Oncology
University of Melbourne
Melbourne, Victoria 3052, Australia

cancers caused by Lynch syndrome occur more frequently and at a younger age in the affected group.[9]

At present, surveillance by regular colonoscopy has been the most effective method to monitor and prevent CRC.[4,10] The current guidelines by National Comprehensive Cancer Network state that aspirin may decrease the risk of Lynch Syndrome but optimal dosage and duration of aspirin therapy are uncertain (https://www2.trikobe.org/nccn/guideline/colorectal/english/genetics_colon.pdf). However, recent data from the Cancer Prevention Project 2 (CAPP2) trial has shown additional clinical benefit to colonoscopy with the use of aspirin as a risk-reducing medicine for CRC in Lynch syndrome individuals.[11] The per-protocol analysis results show that taking 600 mg of enteric-coated aspirin daily for at least 2 years with regular colonoscopy resulted in a significant reduction of cancer development (hazard ratio (HR) of 0.41) compared to placebo.[11] However, intention-to-treat analysis did not show a significant CRC incidence reduction.[11] A follow up report indicated the benefit was still evident 20 years after trial induction, well after the intervention had ceased.[12]

However, despite the breadth of research investigating the use of aspirin for cancer risk reduction in both Lynch syndrome patients and the general population, the mechanism of its effectiveness remains unclear. Studies are yet to uncover the molecular process underpinning the effects of aspirin. Also, the optimal duration of treatment and dose of aspirin that balances benefits and side-effects, is yet to be established. Despite a lack of research on the optimal dosage of aspirin, there is some evidence for a recommendation that all gene carriers should take aspirin daily.[13] The Cancer Prevention Project (CaPP3) study is a randomized controlled trial, which seeks to determine the optimal dose of aspirin required for risk reduction. This trial has randomized 1800 mutation carriers to 100 mg vs. 300 mg vs. 600 mg aspirin daily and spans at least 7 years.

Currently, a significant obstacle for the use of aspirin to improve outcome in people with Lynch syndrome is suboptimal communication about its effects with, and within, affected families. An educational resource to support communication currently does not exist in Australia. A decision aid has been recently published by the National Institute for Health and Care Excellence UK on use of aspirin in people with Lynch Syndrome (https://www.nice.org.uk/guidance/ng151/resources/lynch-syndrome-should-i-take-aspirin-to-reduce-my-chance-of-getting-bowel-cancer-pdf-8834927869). Improving communication and education will be crucial to the reduction of morbidity and mortality in this group as screening and risk management options are becoming increasingly effective.[9,14] A thorough understanding of barriers to communication is integral to the development of such educational resources. Studies have found that information provided by the family doctor as well as educational resources provided to the proband improve information dissemination.[14,15] Access to information and educational resources is crucial in providing accurate information and facilitates the communication process within families.[16] In particular, the benefit of written information has been shown to improve information retention and dissemination.[17,18] Dilzell et al. (2015) found that both relatives and at-risk relatives requested information about Lynch syndrome and cancer risk in the form of a brochure or web link.[14] Leaflets have been proven to be effective and useful resources in the past.[19,20] Harnessing such tools will be necessary to increase the uptake of improved health behaviors in Lynch syndrome carriers. As our understanding of risk-reducing medication and improved surveillance for carriers with Lynch-syndrome-related mutations continues to advance, the ability to bridge the gap through health information, communication, and education becomes ever more important.

This article describes the development and evaluation of an educational leaflet that can be used to support the education of people with Lynch-syndrome-related mutations about the uses of aspirin as risk-reducing medication and its potential side-effects. It is intended that this leaflet will be used as an educational tool to aid decision-making and facilitate discussion between mutation carriers and their health care providers.

## 2. Experimental Section

### 2.1. Leaflet

With contributions from a range of experts on health literacy and Lynch syndrome, an A4 educational leaflet was developed describing the risks and benefits of using aspirin.

The leaflet included a 100-people diagram comparing the lifetime chance of developing bowel cancer without colonoscopy and aspirin to either one of these or both. The 100-people diagram is shown in **Figure 1**. To provide the basis for the risk figures in the diagram, Monte Carlo methods and the available data on the effect of colonoscopy and aspirin on CRC incidence were used, to estimate the cumulative lifetime risk taking into account the age people start taking aspirin. The cumulative risk of first CRC in the absence of colonoscopic surveillance in Lynch syndrome mutation carriers up to age 80 years with any of the four MMR genes mutated, separately for males and females, was based on study by Bonadona et al.[21] It was then assumed that 2–3 yearly colonoscopic surveillance reduced CRC incidence by 61% (Hazard ratio (HR) = 0.387).[10,22–24] The effect of aspirin on CRC incidence was obtained from both published and unpublished CAPP2 trial results. It was assumed that 2–4 years aspirin use reduced CRC cancer incidence in Lynch syndrome mutation carriers with delayed effect as per per-protocol-analysis result (i.e., HR = 1 in 0 to <4 years, HR = 0.41 in 4–10 years)[11] and the effect lasted up to 20 years without attenuation (Finlay Macrae, unpublished). It was also assumed that there was no interaction between colonoscopic surveillance and aspirin as per the CAPP2 trial (i.e., additive effect of aspirin on CRC incidence when assuming all participants were under regular colonoscopic surveillance before starting aspirin chemoprevention and colonoscopic surveillance continues up to age 70 years). Cumulative risk of CRC depended on age and sex as well as aspirin chemoprevention starting age. Therefore, for the purpose of the leaflet, 50-year-old men were chosen and the risk of first CRC up to 80 years of age averaged across any MMR gene mutation as the lifetime bowel cancer risk with or without intervention (i.e., no intervention, colonoscopy only, colonoscopy and aspirin) was presented, assuming these men start aspirin chemoprevention at age 50 years.

Two readability assessment scores—the Flesch-Kincaid readability and the Flesch reading ease score—were used to assess the readability level. The content of the leaflet was planned to be

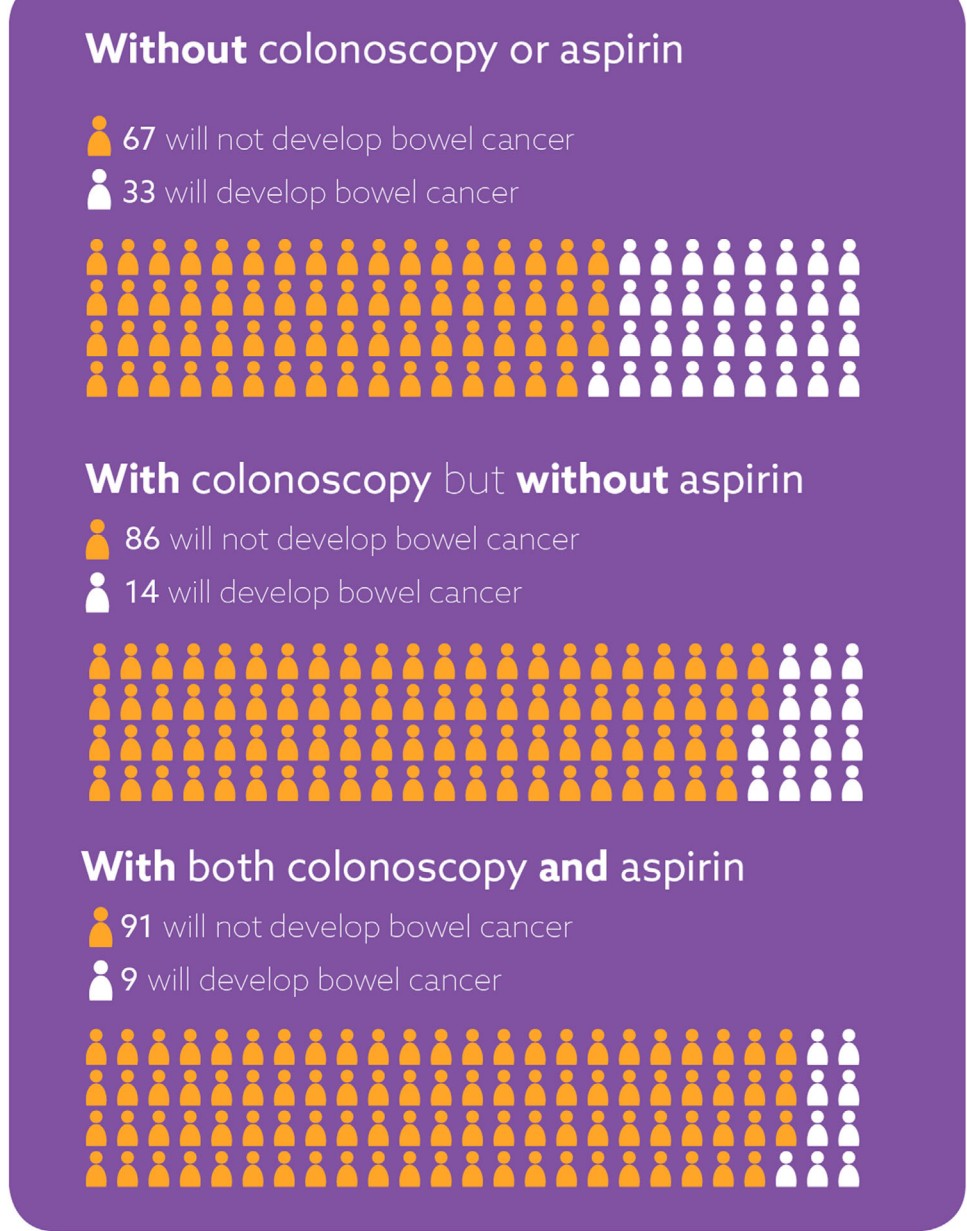

**Figure 1.** 100-people diagram comparing the chance of developing bowel cancer without colonoscopy or aspirin to either with one of these or with both.

at eighth grade reading level, but the leaflet developed had a reading level of twelfth grade.

The ethical approval to test the leaflet was obtained from Human Research Ethics Committee, The Royal Melbourne Hospital, Victoria (HREC/16/MH/354).

### 2.2. Questionnaire

The measures included in the hard copy self-administered questionnaire were adapted from a questionnaire previously developed by this team.[25] It was used to assess the acceptability of content and clarity (9 Likert-type items); satisfaction with the length, relevance, and visual appeal of the resource (7 Likert-type items); as well as the leaflet overall (2 Likert-type items). In addition, open-ended questions were included to enable participants to provide a more detailed evaluation.

### 2.3. Participants

Identified people with Lynch syndrome, aged 18 and over and proficient in English, who had previously attended the Parkville Integrated Familial Cancer Centre, Royal Melbourne Hospital, were invited to participate in the resource evaluation. Information packs, including the leaflet, questionnaire, and a reply-paid

**Table 1.** Participant demographics (*n* = 33).

| Characteristics | |
| --- | --- |
| Age | |
| Mean (years) | 46 |
| Range (years) | 19–75 |
| | *n* [%] |
| Gender | |
| Female | 24 (73) |
| Male | 9 (27) |
| Education | |
| Secondary | 5 (15) |
| Technical and further education | 3 (9) |
| Tertiary | 24 (73) |
| Occupation | |
| Health care | 6 (18) |
| Finance | 3 (8) |
| Education | 4 (12) |
| Student | 2 (6) |
| Retired | 8 (24) |
| Other | 10 (30) |

envelope, were mailed to eligible patients. Participants were asked to opt into the study by reading the leaflet and returning the completed questionnaire. Participant consent was implied if the survey was completed.

### 2.4. Analysis

Descriptive data were generated using the Statistical Package for the Social Science (SPSS) Version 25.0 (SPSS Inc., Chicago, IL).

## 3. Results

In total, 197 Lynch Syndrome carriers were contacted. Thirty-three questionnaires were returned between April 2017 and August 2017. As this was an opt-in survey, the reasons for non-participation are unknown. The demographic characteristics of those who participated are shown in **Table 1**. The sample size did not allow for statistically meaningful comparisons between these sociodemographic groupings. Accordingly, only descriptive statistics such as the mean and mode were used to analyze the results.

## 4. Participants' Views of the Leaflet Overall

In response to Likert-type questions, most responders rated the leaflet as "very" clearly presented (70%), "very" informative (79%), "very" easy to read (64%), and "very" useful (76%). In comparison, participants found the leaflet "very" (45%) or "somewhat" (43%) appealing (**Figure 2**). In addition, 87% of the participants thought the length of the leaflet and the amount of information was "about right."

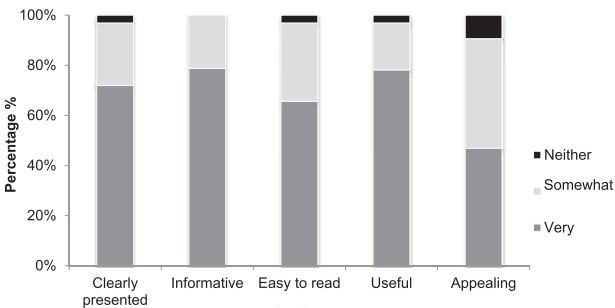

**Figure 2.** Participants' overall view of the leaflet.

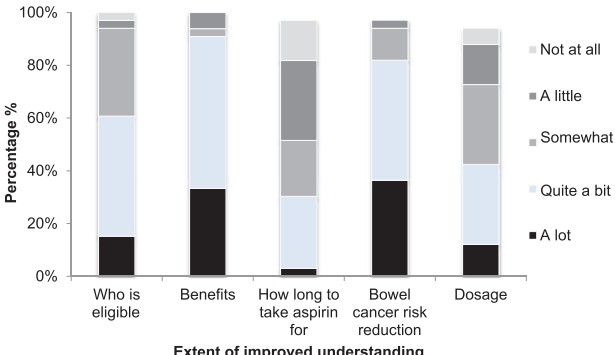

**Figure 3.** Extent of improved understanding.

## 5. Improving Participants' Understanding

Ninety-one percent of participants reported a perceived improvement in their understanding of who is eligible to take aspirin. Of those, 61% reported that their understanding had improved "a lot" or "quite a bit," 33% that it had improved "somewhat" and 3% reported "a little" improvement. With regard to the benefits of aspirin, 100% of participants perceived an improvement in their understanding. Most respondents (91%) felt that their understanding had improved "a lot" or "quite a bit," 3% reported that their understanding had improved "somewhat," and 6% that it had improved "a little" (**Figure 3**).

Similarly, 100% of participants perceived an improvement in their understanding of the bowel cancer risk-reducing effects of aspirin. "Quite a bit" or "a lot" of improvement was reported by 81% of responders, while 12% perceived that their understanding had improved "somewhat" and 3% that it had improved "a little." In relation to how long to take aspirin for, 81% felt that their understanding had improved. Of these only 30% reported "a lot" or "quite a bit" of improvement, while 21% thought their understanding was "somewhat" improved and 30% reported their understanding as "a little" improved. When asked about the dosage of aspirin, 94% perceived an improvement in their understanding. Of these 42% of participants reported "a lot" or "quite a bit" of improvement, 30% rated their understanding as "somewhat" improved and 15% reported "a little" improvement (Figure 3).

Most participants reported some extent of improvement in perceived understanding in knowledge about who is eligible to take aspirin, the benefits of taking aspirin, how long to take aspirin, the reduction in bowel cancer risk from taking aspirin, and the dosage of aspirin. Almost all participants (97%) were satisfied

with the information provided in the leaflet, and 93% found the information in the leaflet relevant. In addition, 80% thought it would be helpful for them to decide about whether to take aspirin, and 60% would recommend it to a friend.

## 6. Response to Open-Ended Questions

Responses to open-ended questions assessing the best and worst aspects of the leaflet, missing or required detail and further suggestions were grouped based on the feature of the leaflet they related to. Participants commented on the photo on the front cover, in relation to which, most participants were satisfied but four participants had negative comments. The participants who gave negative feedback thought the image "could be more culturally inclusive" and some were "really not sure about the smiling faces on the front page."

The 100-person-diagram was mentioned by 14 responders; nine provided positive feedback and five provided negative feedback. All positive feedback about the diagram was given in response to the question about what was "the best part of the leaflet." Three responders specifically favored the use of the image to convey the numerical data describing "the pictures of bowel cancer risk percentages" and "the statistics and the chart" as the best part. One participant "liked" the graph with the people and added "but it also takes up heaps of room" and another thought "the diagrams may not be needed. The explanation seems enough."

The presentation of the information and the content received positive feedback from the majority of responders, who mentioned these aspects in the open-ended questions (11/21). Three responders described the leaflet as "informative" and five thought the leaflet was "easy to read" or "easy to understand." The majority (5/9) of negative comments about the content were in relation to the lack of information about "the unanswered questions" about dosage. In addition, participants were asked about missing information or what parts needed more detail. In response, five of 11 responders requested more information about the dosage of aspirin, and four requested access to research data and additional resources. Finally, six responders suggested changes to the text color, contrast, or size as some of "the writing...is difficult to read."

Changes to the leaflet were made in response to the results of the evaluation. The final version of the leaflet can be found at https://www.psychosocialresearchgroupunsw.org/.

## 7. Discussion

Evaluation of the educational leaflet indicated that the resource was acceptable to most participants. Respondents responded favorably to the overall features of the leaflet including content, length, and visual appeal. In addition, participants reported perceived improvement in the extent of their understanding of the use of aspirin for bowel cancer risk reduction. This leaflet has been designed for dissemination by health care professional to Lynch syndrome carriers as an educational tool to facilitate discussion about using aspirin as a risk-reducing medication. Most respondents were satisfied with the information, found it relevant, and would recommend it to a friend. Importantly, many

participants thought the leaflet would be helpful to them in deciding whether to take aspirin to reduce their risk of developing bowel cancer. Despite the overall acceptance of the leaflet, some participants identified shortcomings of the tool.

The majority of participants who mentioned the 100-people diagram responded positively to its inclusion in this leaflet, underscoring a body of research indicating that pictograms are an effective way to communicate statistical information.[18,26] By contrast, a small proportion of respondents in this study rated the diagram negatively. The responders who criticized the diagram described it as "unnecessary." This may be because the information depicted by the diagram was also provided in the written text. Around one third of those who rated the diagram positively favored the diagram in conjunction with the written explanation, rather than on its own. It has been found that providing information in a number of formats enhances understanding.[18] Given the balance of responses, the diagram was retained given it was deemed as useful in conveying information by most participants.

The content of the leaflet was planned to be at eighth grade reading level but the leaflet developed had a reading level of twelfth grade. The reading level of the leaflet without medical terms such as "Lynch syndrome," "aspirin," and "colonoscopy" reduced the reading level to ninth grade. However, reading comprehension must consider the fact that most of the participants would have been exposed to the term Lynch Syndrome for years, taken aspirin, and even had a colonoscopy themselves. However, despite its high reading level, participants did not comment negatively about the readability of the resource. Most participants provided positive feedback about the content and described the leaflet as "easy to read" and "easy to understand." These participants reported ease in reading this leaflet despite its twelfth grade reading level. Although the reason for this is unclear, it may be related to the education levels of the cohort as the majority had completed tertiary education. Also, given that participants had already attended a familial cancer clinic, they were not naïve about the content covered by the leaflet.

However, many participants made suggestions for improvement of the content of the resource. Responses to open-ended items predominantly highlighted suggestions to have further information included regarding dosage, optimal duration of aspirin treatment, and access to other resources such as journal articles or support groups. Unfortunately, more information about dosage and treatment duration will not be available for several years with the CaPP3 trial anticipated to be completed in 2024. However, reference to relevant journal articles and links to the Lynch syndrome Australia website can be readily included in amendments of this leaflet. Several participants commented on the ease with which the text could be read based on its color and the contrast of the printing. This was an issue for nearly one quarter of all respondents and will require rectification prior to dissemination.

There is a growing body of literature assessing how patients absorb and share information from consultations with their health care professionals.[18,26] Educational resources can be useful for both the patient and the clinician in facilitating understanding and to support decision-making.[14,27] A recent evaluation of aspirin decision aid reported that patients would like to discuss the role of aspirin with their clinician.[28] Another study has shown that some clinicians inform their patients about the

risk-reducing benefits of aspirin, but most would find a brief leaflet useful in their clinical practice when talking to patients about using aspirin as a risk-reducing medicine.[27] Furthermore, as consultations with health professional may become increasingly brief, particularly if many more people with Lynch syndrome are identified through more extensive tumor screening for MMR deficiency, educational resources will become invaluable tools.[27] Not only would an educational resource be useful to improve patients' quality of care,[29] but written information can also improve dissemination of information within the family.[17,30] Our study did not explore if the patients found it easier to discuss the topic with health professionals after reading the leaflet, this should be explored in future studies.

This is the first Australian study to develop and evaluate an educational tool for patients regarding the use of aspirin to reduce bowel cancer risk in people with Lynch syndrome. The growing body of data supporting the use of aspirin as a risk-reducing medicine[11,13,31–33] necessitates communication with those most likely to benefit from these recent findings. In addition, as optimal dosage and treatment duration data will eventually become available, understanding the information needs of this population will be necessary to support the development of future educational tools.

One significant limitation of our study is that the planned 10-year follow up from the CAPP2 reporting significant 35% reduction on the first CRC incidence from the intention-to-treat analysis was published after our study was finished. Therefore, this information was not included in our leaflet and the risk estimate diagrams shown in the leaflet. The effect of colonoscopic surveillance on CRC incidence as reported by Jarvinen et al. could have been an over-estimate, and the cumulative CRC risk varies by MMR genes as reported by the prospective lynch syndrome database (PLSD).

However, the CRC risk by MMR gene reported by the PLSD was from those who were under regular colonoscopic surveillance ranging from 1-yearly to 3-yearly, and currently there is lack of critical information on CRC incidence in pathogenic MMR gene carriers. These include sex- and MMR gene-specific cumulative CRC risk without colonoscopic surveillance, HR associated with colonoscopic surveillance by time since last colonoscopy and the recommended surveillance interval, and the effect of colonoscopy on CRC incidence by MMR gene (i.e., whether CRC risk reduction from colonoscopy differs by MMR gene).

Therefore, in the absence of compelling evidence, we used the average risk and potential protective effect from risk reduction measures across MMR genes and used the conservative estimate on the potential effect of aspirin on CRC risk reduction. Given the purpose of this study was to develop an education resource, we think the simpler message with the conservative estimates will be better suited to communicate with the patients. We are planning to regularly update this educational resource and the updated version will incorporate any emerging evidence in line with the latest clinical recommendations.

This study was a pilot study and these findings have guided improvements made to the educational resource prior to dissemination to patients. The opt-in nature of the evaluation may have led to participation bias, and hence generalizability to the wider population of carriers of Lynch-syndrome-related carriers is somewhat limited. These findings provide suggestions to guide

amendments of this much-needed resource. Furthermore, as more feedback is received as the resource is implemented more widely and as dosage data emerge on publication of the CAPP3 study, this leaflet will form a foundation for the production of future educational tools that provide information about aspirin as a risk-reducing medication.

## 8. Conclusion

This educational leaflet will be a valuable resource to support people with Lynch syndrome in their education and decision-making around taking aspirin as a risk-reducing medicine. Empowering patients to start the conversation with their health professionals and make supported decisions will be an integral first step in bridging the gap from bench to bedside, thus improving health care for people with Lynch syndrome.

## Supporting Information

Supporting Information is available from the Wiley Online Library or from the author.

## Acknowledgements

R.K. and C.M. contributed equally to this work. This research was funded by the Cancer Council New South Wales Strategic Research Partnership (STREP) scheme. B.M. was supported by a Senior Research Fellowship Level B (ID 1078523) from the National Health and Medical Research Council (NHMRC) of Australia. The authors would like to extend their gratitude to the participants involved in this research. B.M. has a remunerated consultant role with the company Astra Zeneca with respect to an unrelated project.

## Conflict of Interest

The authors declare no conflict of interest.

## Data Availability Statement

The data that support the findings of this study are available from the corresponding author upon reasonable request.

## Peer Review

The peer review history for this article is available in the Supporting Information for this article.

## Keywords

aspirin, colorectal cancer, Lynch syndrome, risk reduction

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
