## [**Supplementary Information**: Record of Transparent Peer Review · Advanced Genetics]

Record of Transparent Peer Review

The risk-reducing effect of aspirin in Lynch syndrome carriers: Development and evaluation of an educational leaflet

Rajneesh Kaur*, Cassandra McDonald, Bettina Meiser, Finlay Macrae, Sian K Smith, Yoon Jung Kang, Michael Caruana, Gillian Mitchell

*Corresponding

Review timeline:	Date Submitted:	15-Sep-2021
	Editorial Decision:	18-Oct-2021
	Revision Received:	13-Feb-2022
	Accepted:	14-Feb-2022

Editor: Myles Axton

1st Peer Review

20-Sep to 18-Oct-2021

Reviewer #1

This is a well-designed methodology, and clearly written.

There are some issues which need to be addressed however to help provide context, and limitations need to be addressed

1.1. the data responses are low in this cohort, why was this the case? 33 respondents is rather low, this is a major flaw. The questionnaires were collected over 4 years ago in 2017, would attitudes have changed since then? This needs to be clearly stated in the abstract, and perhaps more respondents are required.

1.2. what is the context in terms of existing guidelines?

1.3. the statement 'This is the first study to develop and evaluate an educational tool for patients regarding the use of aspirin to reduce bowel cancer risk in people with Lynch syndrome.' is not correct. NICE have developed a similar decision aid published in 2020 <https://www.nice.org.uk/guidance/ng151/resources/lynch-syndrome-should-i-take-aspirin-to-reduce-my-chance-of-getting-bowel-cancer-pdf-8834927869>
how do these decision aids compare?

Reviewer #2

This manuscript describes an educational leaflet explaining the purpose and effects of aspirin intake and its evaluation by a group of Lynch syndrome patients. Overall, the tool was favourably accepted by the participants and some suggestions were provided for its improvement. The development of similar tools has proven to be effective for patient education in many areas of medicine, including hereditary cancer and Lynch syndrome. The use of aspirin to reduce cancer risk is a complex topic, since the evidence in favour of its beneficial effects is still somewhat debated.

2.1 Indeed most scientific societies and professional bodies do not fully endorse its use, given the limited available evidence, derived from a single study.

The manuscript is well written and the results are clearly presented and discussed. However, I have the following comments for the authors:

2.2-Information provided on the leaflet about the risk reducing effects is presented as definitive evidence (ie, "Research shows taking aspirin and having a colonoscopy are very effective to lower the chance of getting bowel cancer caused by Lynch syndrome"). This applies to colonoscopy, since there are multiple studies showing its beneficial effects. On the other hand, as mentioned above, there is some debate on the real effectiveness of aspirin in MMR gene pathogenic variant carriers, which is derived from a single, albeit large, study. So, I would suggest to be less definitive about the risk reducing effects of aspirin, ie. by specifying that this evidence is provided by a single study and further confirmation is awaited. This issue should also be discussed more extensively in the manuscript text.

2.3 -It would be interesting to learn whether the patients found it easier to discuss the topic with health professionals (the ultimate goal of this intervention) after reading the leaflet. Was this taken into account in the survey?

2.4 -The number of respondents is low, but anyhow sufficient for a pilot study. Any explanation for the low uptake?

-On p. 4, 2nd paragraph, "...as well as educational resources provided by the proband improve information dissemination." I wonder whether it should be "provided to the proband" rather than "provided by the proband".

Reviewer #3

This is important work given the frequency of Lynch syndrome and the impact of aspirin as a cancer preventive agent. The evaluation and efforts to develop educational material to help address this complex issue is to be welcomed. I would make the following changes to improve the paper.

3.1 While this work was underway, the 10 year planned follow up of CAPP2 was published (and recorded by ASCO as the prevention breakthrough of the year)

Unlike the interim analysis published in 2011, the new paper showed the cancer preventive effect to be apparent on an Intention to Treat basis. For the benefit of patient guidance, it is sensible to use the Incident Rate Ratio, which takes account of all cancers and not just the first, and to refer to the Per protocol analysis where participants took the equivalent of 2 years treatment as originally described. This indicates a 50% reduction in colorectal cancers.

3.2 The paper refers to the Jarvinen paper as evidence of a significant reduction in cancers attributable to colonoscopy. That paper compared people who agreed to surveillance with those who declined but excluded prevalent colorectal cancers in the first group ie. their first CRCs were not counted. This gave a distorted analysis of the effect of colonoscopy. The www.PLSD.eu prospective follow up of more than 8000 LS carriers has not supported this interpretation. Indeed, their analysis in different countries has not confirmed a preventive effect of colonoscopy and suggests that a significant number of cancers arise without an obvious adenoma stage. More frequent examinations were associated with a higher frequency of cancers, possibly due to the discovery of cancers that would be destroyed by immune rejection based on their high visibility to the immune system. The diagram describing cases with colonoscopy only should be revised to reduce the preventive effect of colonoscopy while suggesting these examinations are likely to down stage cancers and enhance long term survival.

3.3 Finally, the prospective database also demonstrates a marked variation in the colon cancer risk with different mismatch repair genes. It is important not to group PMS2 (prospective colorectal cancer risk of 10%) and MSH6 (20%) with the other two (risk 50%).. A gene-specific guide based on the PLSD data would be more compatible with guidance being given by clinical geneticists and expert cancer clinics around the world.

References 11 and 25 are the same paper

1 st Editorial Decision	18-Oct-2021
Editorial decision: Invite resubmission with minor revisions	
Editor's understanding of the reviews	
Reviewer #1 Recommends: Major revision	
Reviewer #2 Recommends: Major revision	
Reviewer #3 Recommends: Minor revision	

Author's Response to 1 st Review	13-Feb-2022
---	-------------

These are the main reviewer recommendations that the editors believe will make the biggest improvement to this article. **Please do address all reviewer comments listed in the decision letter in your point-by-point response** (you may continue this table to do so if you wish). We hope this summary helps you to understand our decision and expedites the revision process. We value feedback from author and referees alike.
AdvGenet@wiley.com

Reviewer comments	Editor recommendation	Author reply	Changes to Manuscript
1.3 NICE have developed a similar decision aid published in 2020...how do these decision aids compare?	ED1 It is essential to introduce similar studies in the Introduction and to compare your study in the Discussion.	We agree and have referenced the NICE's decision aid.	Please see track changes to the introduction.
1.1. the data responses are low in this cohort, why was this the case? 2.4 The number of respondents is low, but anyhow sufficient for a pilot study. Any explanation for the low uptake?	If you cannot add a replication, then explain the reason for the low number of responses and compare the results to other surveys of similar information leaflets.	This was a pilot test and therefore the sample size was deemed sufficient. We have added a recommendation to explore patients' attitudes with a bigger sample size. The data was collected by familial cancer care heads, and it was hard for them to carry on recruitment for longer term due to other commitments.	Please see the track changes in the discussion section.
1.2. what is the context in terms of existing guidelines?	ED2 Discuss the leaflet and results relative to guidelines available in	We have referred to NCCN guidelines 2020.	Please see the track changes to the introduction section.

2.1 Indeed most scientific societies and professional bodies do not fully endorse its [aspirin] use	2017 and guidelines now published.		
2.3 It would be interesting to learn whether the patients found it easier to discuss the topic with health professionals (the ultimate goal of this intervention) after reading the leaflet. Was this taken into account in the survey?	ED3 If you cannot get more breadth and power to the study, perhaps report greater depth: did more people discuss their options with a healthcare provider in this study or that of NICE?	We did not explore it in our study but has added it as a recommendation for future studies.	Please see the track changes in the discussion section.
3.1 For the benefit of patient guidance, it is sensible to use the Incident Rate Ratio, which takes account of all cancers and not just the first, and to refer to the Per protocol analysis where participants took the equivalent of 2 years treatment as originally described. This indicates a 50% reduction in colorectal cancers.	ED4 Use this ratio and cite the CAPP2 trial as source.		
3.2 The diagram describing cases with colonoscopy only should be revised to reduce the preventive effect of colonoscopy while suggesting these examinations are likely to down stage cancers and enhance long term survival. 3.3 It is important not to group PMS2 (prospective colorectal cancer risk of 10%) and MSH6 (20%) with the other two (risk 50%).. A gene-specific guide based on the PLSD data would be more compatible with guidance being given by clinical geneticists and expert cancer clinics around the world.	ED5 revise the diagram and cite the PLSD.eu prospective study as source. Ensure that your discussion reflects the recommendation of this study to take gene-specific risk into account. Discuss whether this distinction is relevant to public outreach about aspirin prevention.		

Reviewer #1

This is a well-designed methodology, and clearly written.

There are some issues which need to be addressed however to help provide context, and limitations need to be addressed

1.1. the data responses are low in this cohort, why was this the case? 33 respondents is rather low, this is a major flaw. The questionnaires were collected over 4 years ago in 2017, would attitudes have changed since then? This needs to be clearly stated in the abstract, and perhaps more respondents are required.

Response: The sample size considerations are fine as this a pilot test and for a pilot test a sample size of 10-40 is considered okay. We have added to the abstract that the leaflet was pilot tested in 2017 and have also added further exploration of attitudes in the overall conclusion of the study.

1.2. what is the context in terms of existing guidelines?

Response: The existing guidelines are not definitive in terms of taking aspirin for risk reduction as the evidence is evolving. We have referred to NCCN's guidelines in the introduction.

1.3. the statement 'This is the first study to develop and evaluate an educational tool for patients regarding the use of aspirin to reduce bowel cancer risk in people with Lynch syndrome.' is not correct. NICE have developed a similar decision aid published in 2020 <https://www.nice.org.uk/guidance/ng151/resources/lynch-syndrome-should-i-take-aspirin-to-reduce-my-chance-of-getting-bowel-cancer-pdf-8834927869>

how do these decision aids compare?

Response: We have referred to NICE decision aid and have changed it to first Australian study.

Reviewer #2

This manuscript describes an educational leaflet explaining the purpose and effects of aspirin intake and its evaluation by a group of Lynch syndrome patients. Overall, the tool was favourably accepted by the participants and some suggestions were provided for its improvement. The development of similar tools has proven to be effective for patient education in many areas of medicine, including hereditary cancer and Lynch syndrome. The use of aspirin to reduce cancer risk is a complex topic, since the evidence in favour of its beneficial effects is still somewhat debated.

2.1 Indeed most scientific societies and professional bodies do not fully endorse its use, given the limited available evidence, derived from a single study.

The manuscript is well written and the results are clearly presented and discussed. However, I have the following comments for the authors:

2.2-Information provided on the leaflet about the risk reducing effects is presented as definitive evidence (ie, "Research shows taking aspirin and having a colonoscopy are very effective to lower the chance of getting bowel cancer caused by Lynch syndrome"). This applies to colonoscopy, since there are multiple studies showing its beneficial effects. On the other hand, as mentioned above, there is some debate on the real effectiveness of aspirin in MMR gene pathogenic variant carriers, which is derived from a single, albeit large, study. So, I would suggest to be less definitive about the risk reducing effects of aspirin, ie. by specifying that this evidence is provided by a single study and further confirmation is awaited. This issue should also be discussed more extensively in the manuscript text.

Response: Given the current guidelines and evidence presented, we have used less definitive terms.

2.3 -It would be interesting to learn whether the patients found it easier to discuss the topic with health professionals (the ultimate goal of this intervention) after reading the leaflet. Was this taken into account in the survey?

Response: This was not asked in our survey and we have acknowledged it and added as a recommendation for the future studies.

2.4 -The number of respondents is low, but anyhow sufficient for a pilot study. Any explanation for the low uptake?

-On p. 4, 2nd paragraph, "...as well as educational resources provided by the proband improve information dissemination." I wonder whether it should be "provided to the proband" rather than "provided by the proband".

Response: We have made this correction.

Reviewer #3

This is important work given the frequency of Lynch syndrome and the impact of aspirin as a cancer preventive agent. The evaluation and efforts to develop educational material to help address this complex issue is to be welcomed. I would make the following changes to improve the paper.

3.1 While this work was underway, the 10 year planned follow up of CAPP2 was published (and recorded by ASCO as the prevention breakthrough of the year)

Unlike the interim analysis published in 2011, the new paper showed the cancer preventive effect to be apparent on an Intention to Treat basis. For the benefit of patient guidance, it is sensible to use the Incident Rate Ratio, which takes account of all cancers and not just the first, and to refer to the Per protocol analysis where participants took the equivalent of 2 years treatment as originally described. This indicates a 50% reduction in colorectal cancers.

3.2 The paper refers to the Jarvinen paper as evidence of a significant reduction in cancers attributable to colonoscopy. That paper compared people who agreed to surveillance with those who declined but excluded prevalent colorectal cancers in the first group ie. their first CRCs were not counted. This gave a distorted analysis of the effect of colonoscopy. The www.PLSD.eu prospective follow up of more than 8000 LS carriers has not supported this interpretation. Indeed, their analysis in different countries has not confirmed a preventive effect of colonoscopy and suggests that a significant number of cancers arise without an obvious adenoma stage. More frequent examinations were associated with a higher frequency of cancers, possibly due to the discovery of cancers that would be destroyed by immune rejection based on their high visibility to the immune system. The diagram describing cases with colonoscopy only should be revised to reduce the preventive effect of colonoscopy while suggesting these examinations are likely to down stage cancers and enhance long term survival.

3.3 Finally, the prospective database also demonstrates a marked variation in the colon cancer risk with different mismatch repair genes. It is important not to group PMS2 (prospective colorectal cancer risk of 10%) and MSH6 (20%) with the other two (risk 50%).. A gene-specific guide based on the PLSD data would be more compatible with guidance being given by clinical geneticists and expert cancer clinics around the world.

Response to 3.1, 3.2 and 3.3: As per the reviewer's comment, the planned 10-year follow up from the CAPP2 reporting significant 35% reduction on the first colorectal cancer (CRC) incidence from the intention-to-treat analysis was published after our study was finished. We also agree with the reviewer that the effect of colonoscopic surveillance on CRC incidence as reported by Jarvinen et al. could have been an over-estimate, and the cumulative CRC risk varies by mismatch repair (MMR) genes as reported by the Prospective Lynch Syndrome Database (PLSD). However, the CRC risk by MMR gene reported by the PLSD was from those who were under regular colonoscopic surveillance ranging from 1-yearly to 3-yearly, and there currently is lack of critical information on CRC incidence in pathogenic MMR gene carriers. These include sex- and MMR gene-specific cumulative CRC risk without colonoscopic surveillance, hazard ratio (HR) associated with colonoscopic surveillance by time since last colonoscopy and the recommended surveillance interval, and the effect of colonoscopy on CRC incidence by MMR gene (i.e., whether CRC risk reduction from colonoscopy differs by MMR gene).

Therefore, in the absence of compelling evidence, we used the average risk and potential protective effect from risk reduction measures across MMR genes and used the conservative estimate on the potential effect of aspirin on CRC risk reduction (this is in line with Reviewer 2's comment, the effect of aspirin on CRC incidence is from one RCT). Given the purpose of this study was to develop an education resource, we think the simpler message with the conservative estimates will be better suited to communicate with the patients. We are planning to regularly update this educational resource and the updated version will incorporate any emerging evidence in line with the latest clinical recommendations.

References 11 and 25 are the same paper

Response: We have corrected this.

Final Decision	14-Feb-2022
----------------	-------------

The authors have responded to the reviewers' recommendations and the article is accepted for publication.